# *Sensortoolkit*—A Python Library for Standardizing the Ingestion, Analysis, and Reporting of Air Sensor Data for Performance Evaluation

**DOI:** 10.3390/s25185645

**Published:** 2025-09-10

**Authors:** Menaka Kumar, Samuel G. Frederick, Karoline K. Barkjohn, Andrea L. Clements

**Affiliations:** 1Former Oak Ridge Associated Universities (ORAU), Contractor to the U.S. Environmental Protection Agency Through the National Student Services Contract, 100 ORAU Way, Oak Ridge, TN 37830, USA; kumar.menaka@epa.gov (M.K.); sf20@illinois.edu (S.G.F.); 2Office of Research and Development, U.S. Environmental Protection Agency, 109 T.W. Alexander Way, Research Triangle Park, Durham, NC 27711, USA; barkjohn.karoline@epa.gov

**Keywords:** air sensor, collocation, performance, open-source software, Python, accuracy, evaluation, data analysis, data visualization

## Abstract

**Highlights:**

**What are the main findings?**
The U.S. EPA previously released a series of reports with recommendations to standardizing the summary and presentation of air sensor performance evaluations through collocation with federal reference and equivalent method instruments including recommended statistical metrics and figures.The U.S. EPA is introducing the free and open-source Python library called *sensortoolkit* for the analysis of air sensor performance evaluation data which handles a wide variety of data formats, calculates the metrics, and creates summary figures.

**What is the implication of the main finding?**
The library will reduce the data processing effort and support the standardization of air sensor performance evaluation results.Consistency in reporting test results will help consumers compare performance results and make more informed purchasing decision.

**Abstract:**

Open-source software tools designed specifically for evaluating and reporting air sensor performance are limited. The available tools do not provide a means for summarizing the sensor performance using common statistical metrics and figures, nor are they suited for handling the wide variety of data formats currently used by air sensors. We developed *sensortoolkit* v1.1.0 as a free, open-source Python v3.8.20 library to encourage the use of the U.S. Environmental Protection Agency’s (U.S. EPA) recommended performance evaluation protocols for air sensors measuring particulate matter and gases. The library compares the collocated air sensor against reference monitor data and includes procedures to reformat both datasets into a standardized format using an interactive setup module. Library modules calculate performance metrics (e.g., the coefficient of determination (R^2^), slope, intercept, and root mean square error (RMSE)) and make plots to visualize the data. These metrics and plots can be used to better understand sensor accuracy, the precision between sensors of the same make and model, and the influence of meteorological parameters at 1 h and 24 h averages. The results can be compiled into a reporting template allowing for the easier comparison of sensor performance results generated by different organizations. This paper provides a summary of the *sensortoolkit* and a case study to demonstrate its utility.

## 1. Introduction

Air sensors have undergone a rapid adoption, leading to a broader access to insightful air quality data. Air sensors have been used in numerous non-regulatory, supplemental, and informational monitoring applications. Some examples include community-wide network deployments which increase the spatial density and temporal resolution of local air quality measurements [1,2,3,4] and educational programs aimed at increasing the community awareness of local air pollution sources and air quality trends [5,6,7,8]. The use of air sensors has led to collaborative efforts among stakeholders such as community members and tribal, local, state, and federal representatives [9,10,11]. The heightened public awareness and use of air sensors have allowed for greater information exchange and the awareness of air quality issues and concerns.

Despite the significant transformation air sensors have facilitated in altering how monitoring efforts are undertaken, it is well-established that the data quality of air sensors varies widely [12,13,14]. Most, but not all, particulate matter (PM) sensors capture the trends in the variation in fine particulate matter (PM_2.5_) concentrations but fewer accurately capture the variation in coarse particulate matter concentrations (PM_c_ or PM_10-2.5_) [15,16,17]. PM and gas sensors may be less accurate and under- or over-report concentrations relative to the collocated Federal Reference Method (FRM) or Federal Equivalent Method (FEM) instruments requiring bias adjustment or correction and extensive quality assurance or data cleaning [12,18]. The data may also be influenced by the ambient relative humidity or other meteorological conditions [19,20,21,22]. Sensors measuring gaseous pollutants such as ozone (O_3_), nitrogen dioxide (NO_2_), sulfur dioxide (SO_2_), and carbon monoxide (CO) often experience interference from other pollutants [15,22,23,24]. The wide variation in performance and the need to understand sensor data quality, and all the potential conditions that may impact the quality, make performance evaluation essential.

Air sensors are not subject to the rigorous standards for measurement accuracy and precision that regulatory-grade instruments must meet. In the United States, the U.S. Environmental Protection Agency (U.S. EPA) designates candidate methods as either FRMs or FEMs following extensive testing to ensure compliance with performance standards set by the Agency (i.e., 50 CFR part 53). Air sensor performance evaluations are conducted by collocating devices alongside FRM/FEM instruments to quantify the accuracy, precision, and bias of sensor measurements and to understand environmental influences. Although this strategy is common practice today, it is difficult to compare evaluations due to differences in how performance evaluations are conducted and how the results are reported.

Since 2021, the U.S. EPA has released a series of reports outlining performance testing protocols, metrics, and target values for sensors measuring PM_2.5_ [21], O_3_ [22], particles with diameters that are generally less than 10 μm (PM_10_) [25], and NO_2_, SO_2_, and CO [23]. These reports, subsequently referred to as Sensor Targets Reports, provide recommended procedures for testing sensors used in ambient, outdoor, fixed-site settings. These procedures are designed to help us understand sensor performance “out of the box” and as designed by the manufacturer to give a prospective user an idea of how that sensor will perform when purchased. Testing is divided into two phases, including the “base testing” of sensors at an ambient monitoring site and “enhanced testing” in a laboratory chamber. During each phase, sensors are collocated alongside an FRM or FEM instrument. For base testing, the Sensor Targets Reports recommend evaluating sensor performance against a suite of performance metrics and associated target values and provides details about how each metric should be calculated. Testers are encouraged to compile the base and enhanced testing results, including figures and tabular statistics, into a reporting template. The template complements the goals of the Sensor Targets Reports by offering a common framework for displaying the sensor performance evaluation results.

The lack of uniform reporting of sensor data is a significant challenge. For example, two different sensors measuring the same pollutant might use different comment and header formats, data reporting frequencies, data averaging methodologies, timestamp formats, pollutant/parameter nomenclature, delimiters, and/or file types. They may measure different parameters or a different number of parameters and use variable definitions for similarly named parameters. Metadata to assist in data interpretation may be embedded within the file, or in a separate file, or may be missing. As a result, individuals intending to evaluate air sensor data may need to develop a custom code for the data analysis, including functions for data ingestion and averaging. Such code may not be extensible to numerous sensor data formats or pollutants, requiring individuals to create sensor-specific code. This fragmented approach requires extensive coding knowledge in order to analyze sensor data, is time-intensive, and complicates the accessibility of air sensor use and performance.

Many organizations have proposed solutions in response to the need for software tools that allow the analysis and visualization of sensor data. However, these tools are limited in scope and/or accessibility. Some sensor manufacturers offer an online platform for viewing sensor measurements, status, and, occasionally, the ability to download data via an application programming interface (API). These platforms may be provided using a “software as a service” business model, whereby the sensor user pays for a subscription giving them continued access to the platform. Such platforms may prove costly for users operating under a limited budget. Other tools for analyzing sensor data are provided as downloadable software packages. These packages typically comprise modules and functions, written in one or multiple coding languages, to allow users to import sensor data on the user’s system or acquire data from an API and generate various statistical values and figures. Software packages are commonly provided as free and open-source software and may be built on open-source programming languages such as R or Python. Existing open-source packages for analyzing air sensor data may be limited to a single or small subset of sensor types. The AirSensor v1.1.1 R package developed by the South Coast Air Quality Management District (South Coast AQMD) and Mazama Science is an example providing an extensive set of tools for analyzing the data collected by the PurpleAir PA-II [26,27]. Other software packages, such as OpenAir v2.19.0 [28], allow for a broader utilization of air quality data. However, these may not be specifically tailored to air sensor data and, thus, lack important in-package utilities for evaluating the air sensor performance against regulatory-grade data.

Here, we introduce the free and open-source Python library called *sensortoolkit* v1.1.0 for the analysis of air sensor data. The *sensortoolkit* library allows for (1) ingesting and importing sensor and reference data from a variety of data formats into a consistent and standardized formatting scheme, (2) the time averaging and analysis of sensor data using statistical metrics and target values recommended by the Sensor Targets Reports, as well as the visualization of sensor data trends via scatterplots, time-series graphs, etc., and (3) compiling performance evaluation results into the standardized reporting template provided in the Sensor Targets Reports. The code library, documentation, and example datasets, which can be used to learn how to use the library, are available on GitHub [29]. A case study of the *sensortoolkit* will be presented using air sensor data from Phoenix, AZ, along with an interpretation of the performance evaluation results.

## 2. Materials and Methods

### 2.1. Suggested User Experience

The *sensortoolkit* library is most suitable for individuals who have some prior Python coding experience. The library is equipped with an API that allows for ease of navigation and selection of library modules and methods. The library’s functionality is mediated by a user-friendly object-oriented approach. This streamlines the need for user-input while allowing for reliable interoperability between *sensortoolkit* subroutines.

### 2.2. Required Software

The *sensortoolkit* package is free and open-source and is compatible with MacOS and Windows. It was developed via an Anaconda distribution of Python version 3.8.8 [30] and can be used with Python version 3.6+. While users are not required to download an Anaconda distribution to use *sensortoolkit*, the Python distribution contains numerous Python packages that are utilized by *sensortoolkit*
https://github.com/USEPA/sensortoolkit/network/dependencies (accessed on 27 November 2024). Software utilities such as the conda package and environment management system and the Spyder integrated development environment (IDE) for scripting, compiling, and executing code (2016) may be helpful. *Sensortoolkit* is hosted and distributed via the Python Packaging Index (PyPI) (https://pypi.org/project/sensortoolkit/ (accessed on 6 November 2024)) and available from the U.S. EPA public GitHub repository [29]. U.S. EPA does not intend to maintain or continue development of this package, but users can further develop and expand on its features.

### 2.3. Documentation

Documentation for *sensortoolkit* is built as html using Sphinx (https://www.sphinx-doc.org/en/master/ (accessed on 6 November 2024)) and deployed to ReadtheDocs (https://sensortoolkit.readthedocs.io/en/latest/ (accessed on 28 August 2025)). This documentation provides an overview of the *sensortoolkit* including its installation and use, component data structures, and data formatting scheme. The API documentation provides a comprehensive description of sub-packages, modules, and methods contained within *sensortoolkit*.

### 2.4. Design and Architecture

The *sensortoolkit* package is designed to be highly extensible to a wide range of air sensor datasets and pollutants in a procedurally consistent and reproducible manner. This is achieved by organizing workflows associated with *sensortoolkit* around an object-oriented approach. The objects and general flow of *sensortoolkit* are shown by Figure 1.

To get started, users can find an initial setup script in the online documentation (https://sensortoolkit.readthedocs.io/en/latest/template.html#script-template (accessed on 4 December 2024)). This script provides a line-by-line example for how to utilize *sensortoolkit* to conduct sensor evaluations and is intended to be executed code-block by code-block. It includes various components to set the path, create directories, collect information about the testing organization and testing location, specify the sensor make/model, run the sensor and reference setup routines (Section 2.4.1), and specify what pollutant report the user wishes to create. During this process, the user supplies information about the testing organization including the name of the test, organization conducting the test (e.g., name, division, and type of organization), and contact details (e.g., website, email, and phone number). The user can also supply information about the test location including site name, address, latitude, longitude, and air monitoring station’s Air Quality System (AQS) Identification Number (ID), if applicable.

The sensor and reference setup routines (Section 2.4.1) prompt users to supply sensor data in .csv, .txt, and/or .xlsx format and reference monitor data either as a file or by pulling in reference data using AQS, AirNow, or AirNow-Tech APIs and the user’s existing authentication key. Using a series of prompts to describe the data being supplied, a set of three testing attribute objects (sensortoolkit.AirSensor, sensortoolkit.ReferenceMonitor, and sensortoolkit.Parameter) are created and specified for the air sensor, reference monitor, and pollutant to be analyzed, respectively. The testing attribute objects are then passed to the evaluation objects (sensortoolkit.SensorEvaluation and sensortoolkit.PerformanceReport) and the code calculates performance metrics and/or produces a testing report. A detailed discussion of testing attribute objects and evaluation objects is summarized in Section 2.4.1 and Section 2.4.2, respectively. Subsequent mentions of either testing attribute objects or evaluation objects will omit the use of “sensortoolkit.[object]” terminology, indicating it is a *sensortoolkit* package object.

#### 2.4.1. Testing Attribute Objects

Testing attribute objects are intended to house key data and information necessary for conducting sensor performance evaluations. These include AirSensor, which contains 1 h- and 24 h-averaged datasets processed into a standardized format, alongside basic sensor information such as the sensor make, model, and optional parameters including manufacturer-stated operational ranges for temperature and relative humidity and the firmware version at the time datasets were recorded. The ReferenceMonitor testing attribute object contains 1 h- and 24 h-averaged data for reference monitor datasets acquired via API (AirNow, AQS, and AirNow-Tech) or from locally stored datasets. The third testing attribute object, the Parameter object, contains information about the pollutant for which the air sensor performance will be evaluated. Parameter objects can be created for common pollutants within the library’s data formatting scheme (Section 2.4.2). Parameter objects contain numerous attributes such as reference measurement averaging intervals, whether the parameter is a criteria pollutant, and its AQS parameter code, as well as units of measure and the corresponding AQS unit code.

Users who have not previously created a setup configuration file for the air sensor they intend to evaluate will start with the AirSensor.sensor_setup() method. This method was designed to be sensor-agnostic and flexible to work with a wide variety of sensor data formats and file types. A series of prompts asks users to describe the data files (Figure 2) so that the data can be properly identified, labeled, and, ultimately, transformed into the standard data formatting scheme described in Section 2.4.2. First, users are prompted to specify the sensor make and model, file type (.csv, .txt, or .xlsx), and file location. Once files are imported, users may use the ingestion module prompts to specify the location of the header row (if present) and to describe each column within the file indicating what data is included, the concentration and meteorological units when applicable, and the format of the timestamp. Prompts also ask the user to specify the time zone and a unique serial or ID number for each sensor. The setup module will use this information to create a setup.json file.

Users follow a similar procedure to start the ReferenceMonitor object reference_setup() module which creates and configures the ReferenceMonitor object. As noted previously, users may query reference monitor data from the AQS API (https://aqs.epa.gov/aqsweb/documents/data_api.html (accessed on 1 September 2025)). This method is preferred because the stored data has undergone rigorous quality control. There can be a delay in data availability as data is typically quality-assured and uploaded quarterly. Data can also be pulled from the AirNow API (https://docs.airnowapi.org/ (accessed on 1 September 2025)) which provides real-time air quality data from monitors managed by state, tribal, local, and federal agencies. However, data from AirNow have not been as extensively validated and verified in the same manner as AQS data. Finally, data can also be pulled from AirNow-Tech but must be manually downloaded to the user’s computer from the AirNow-Tech website (https://airnowtech.org/ (accessed on 1 September 2025)) prior to use with *sensortoolkit*. Unpivoted datasets from AirNow-Tech are preferred by *sensortoolkit*.

To reanalyze a sensor dataset at a later date with the same testing attribute objects, users can use the setup.json configuration files saved for the AirSensor and ReferenceMonitor objects, eliminating the need to go through all of the setup prompts again. This idea also applies to new sensor evaluations that match sensor types that have been previously evaluated. With these files, users can import sensor and reference data and process files that have been previously converted to the *sensortoolkit* data formatting scheme (SDFS) (Section 2.4.2). The json file is saved in the user’s/data folder within their project directory.

Depending on the temperature unit (i.e., °F or °C) provided in the file and specified during setup, the SensorSetup and/or ReferenceSetup objects will determine if a conversion is required. If so, the ingestion module will undertake that conversion. Plots created by the SensorEvaluation object are presented in degrees Centigrade.

Following setup, the AirSensor and ReferenceMonitor testing attribute objects contain information about the air sensor (AirSensor), reference monitor (ReferenceMonitor), and the pollutant or environmental parameter of interest for the evaluation (Parameter). The AirSensor and ReferenceMonitor objects house time-series datasets at the original recorded sampling frequency and at averaging intervals specified by the Parameter object (e.g., PM_2.5_ data are averaged to both 1 h and 24 h averages, and O_3_ data are averaged to 1 h intervals). The averaging intervals are specified in accordance with the sampling and averaging intervals reported by FRMs and FEMs for the pollutant of interest and are not user-specified. Both objects also store device attributes (e.g., make and model). Testing attribute objects (Section 2.4.1) are passed on to the two evaluation objects (Section 2.4.3 and Section 2.4.4) to compute various quantities and metrics, create plots, and compile reports as recommended by Sensor Targets Reports [21,22,23,25].

#### 2.4.2. Data Formatting Scheme

Converting both sensor and reference datasets into a common formatting standard allows for ease of use in accessing and analyzing these datasets. The SDFS is used by the library for storing, cataloging, and displaying collocated datasets for air sensor and reference monitor measurements. SDFS versions of sensor and reference datasets are created automatically following completion of the respective setup modules for the AirSensor and ReferenceMonitor objects. Appendix A details the parameter labels and descriptions used in the SDFS. The list of parameters contains common pollutants including both particulates and gas phase compounds and many are deemed criteria pollutants by U.S. EPA. The SDFS groups columns in sensor datasets by the parameter, with parameter values in one column and the corresponding units in the adjacent column.

SDFS is intended for use with time-series datasets recorded via continuous monitoring at a configuring sampling frequency, whereby a timestamp is logged for each consecutive measurement. Each row entry in SDFS datasets is indexed by a unique timestamp in ascending format (i.e., the head of datasets contain the oldest entries, while the tail contains the most recently recorded entries). Timestamps are logged under the “DateTime” column, and entries are formatted using the ISO 8601 extended format. Coordinated Universal Time (UTC) offsets are expressed in the form ±[hh]:[mm]. For example, the timestamp corresponding to the last second of 2021 would be expressed as “2021-12-31T11:59:59+00:00” in the Greenwich Mean Time (GMT) Zone. By default, datasets are presented and saved with timestamps aligned to UTC.

#### 2.4.3. Sensor Evaluation Object

The SensorEvaluation object uses instructions from the Sensor Targets Reports to calculate performance metrics and generate figures and summary statistics.

The SensorEvaluation.calculate_metrics() method creates data structures containing tabular statistics and summaries of the evaluation conditions. A pandas DataFrame is created which contains linear regression statistics for the sensor vs. FRM/FEM comparison. The linear regression statistics included are linearity (R^2^), bias (slope and intercept), RMSE, N (number of sensor–FRM/FEM data point pairs), and precision metrics (coefficient of variation (CV) and standard deviation (SD)). These statistics describe the comparability of the tested sensors, as well as the minimum, maximum, and the average pollutant concentrations. Data are presented for all averaging intervals specified by the Parameter object. A deployment dictionary, SensorEvaluation.deploy_dict, is also created which contains descriptive statistics, sensor uptime at 1 and 24 h averaging intervals, and textual information about the deployment, including the details about the testing agency and site supplied by the user during setup. This information is written and saved to a json file and is accessed when creating the final report.

The SensorEvaluation object also creates several plots. These include time-series plots, sensor vs. reference scatterplots, performance metric plots, frequency plots of meteorological conditions, and figures exploring how meteorological conditions impact sensor performance.

The SensorEvaluation_plot_timeseries() method plots display sensor and reference concentration measurements along the y-axis as a function of time on the x-axis. Data are presented for the time-averaging intervals specified by the Parameter object. For particulate matter, one plot is created for 1 h averaging intervals and another for 24 h averaging intervals. For gases, one plot is created for 1 h averaging intervals. The reference measurements are shown in solid black lines and sensor measurements are shown in colored lines with a different color representing each of the collocated sensors. The time-averaging interval, sensor name, and pollutant of interest are displayed in the figure header. These graphs assist users in determining whether the sensor is capturing the trends, or changes in concentrations, similarly to the reference monitor. Users may also be able to identify time periods or conditions when the sensor under- or over-reports concentrations, if the sensor may be prone to producing outliers, and/or when data completeness was poor.

The SensorEvaluation.plot_sensor_scatter() method plots time-matched measurement concentration pairs with sensor measurements along the y-axis and reference measurements along the x-axis. The one-to-one line (indicating ideal agreement between sensor and reference measurements) is shown as a dashed gray line. Measurement pairs are colored by relative humidity measured by the independent meteorological instrument at the monitoring site or *sensortoolkit* will attempt to use onboard sensor relative humidity (RH) measurements. The comparison statistics, linear regression equation, R^2^, RMSE, and N are printed in the upper left of the graphs. Separate plots are created for each collocated sensor. The sensor name, time-averaging interval, and pollutant of interest are displayed in the figure header. These graphs assist users in determining whether the sensor typically under- or over-reports concentrations and whether the sensor/reference comparison is consistent through the testing period. Users can also compare the graphs for each sensor to determine if the sensor/reference comparison is consistent among the three identical sensors.

The SensorEvaluation.plot_metrics() method creates a plot that visualizes how the performance metrics for the sensor compares to the target values proposed in Sensor Targets Reports. This plot is a series of adjacent subplots, each corresponding to a different metric. Ideal performance values are indicated by a dark gray line across the plot, although RMSE, normalized RMSE (NRMSE), CV, and SD would ideally all be zero, making this line difficult to see. Target ranges are indicated by gray shaded regions. The calculated metric for each sensor (R^2^, slope, and intercept) or the ensemble of sensors (RMSE, NRMSE, CV, and SD) are shown as small circles. Metrics are calculated for each of the time-averaging intervals specified in the Parameter object. These graphs assist users in quickly determining which target values are met, how close the sensors are to meeting the missed targets, if there is variation among the batch of sensors tested, and if time-averaging interval affects the sensor’s ability to meet the targets.

The SensorEvaluation.plot_met_dist() method plots the relative frequency of meteorological measurements recorded during the testing period in a bar graph. Temperature (T) and relative humidity (RH) data reflects measurements by the independent meteorological instrument at the monitoring site. The time-averaging interval is displayed in the figure header. These graphs assist users in determining whether the test conditions are similar to conditions in the intended deployment location. If deployment conditions are likely to be significantly different, further testing is advised before deployment.

The SensorEvaluation.plot_met_influence() method creates plots to visualize the influence of T or RH on sensor measurements. Sensor concentration measurements are normalized by (i.e., divided by) the reference concentration measurements and then plotted on the y-axis. T and RH, as measured by the independent meteorological instrument at the monitoring site, are plotted on the x-axis. These graphs assist users in determining whether T and RH influence the performance of the sensor. For instance, if the RH graph shows an upward trend, it would indicate that the sensor concentration measurements are higher during high humidity conditions. Thus, users may wish to include a RH component when developing a correction equation for the sensor measurements.

The SensorEvaluation.plot_wind_influence() method is an optional plot to visualize the influence of wind speed and direction on sensor measurements. For wind speed, sensor concentration measurements are shown as the concentration difference between the sensor and reference monitor and plotted on the y-axis. For wind direction, concentration measurements reported by the sensor are shown as the absolute value of the concentration difference between the sensor and reference monitor and plotted on the y-axis. These graphs assist users in determining whether wind speed and wind direction influence the performance of the sensor.

In addition to plots, the SensorEvaluation object can print descriptive summaries to the console. The SensorEvaluation.print_eval_metrics() method prints a summary of the performance metrics along with the average value and the range (min to max) for that metric for each of the collocated sensors. The SensorEvaluation.print_eval_conditions() method prints a summary of the mean (min to max) pollutant concentration as reported by both the sensor and reference monitor and meteorological conditions (T and RH) experienced during the testing period. These summaries are displayed for each of the time-averaging intervals specified in the Parameter object.

#### 2.4.4. Performance Report Object

The Performance Report object takes the output from the SensorEvaluation object to create a testing report utilizing the Sensor Targets Report reporting template. Information about the testing organization and testing site, specified in the initial setup script, are included at the top of the first page of the testing report. Details about the sensor and FRM/FEM reference instrumentation will be populated into the report from the information obtained in the sensor and reference data ingestion module but can also be adjusted or added manually after the report is created.

Several plots generated via the SensorEvaluation object are displayed on the first page including the time-series plots, one scatterplot, performance metrics and target value plots, the T and RH frequency plots describing test conditions, and plots describing meteorological influences on the sensor data. One goal of the reporting template is to provide as much at-a-glance information as possible on this first page.

The second page of the report is populated with a tabular summary of the performance metrics including R^2^, slope, and intercept for each of the collocated sensors. The tabular summary makes it easier to compare the metrics with the performance targets. Scatterplots for each collocated sensor are displayed on this page or the subsequent page depending on space. Multiple scatterplots are presented here to allow users to visually compare performance among the cadre of sensors.

The subsequent pages of the reporting template are designed to document supplemental information, and testers can begin populating this material after the report is generated. In accordance with the requested information in the Sensor Targets Reports, a large table of supplemental information is printed. Testers can use the check boxes to note if information is available and can manually enter details or URLs as available. If testers require more space or choose to provide more information, additional information can be added to the end of the report.

## 3. Results

As a case study, the *sensortoolkit* Python code library was used to evaluate the performance of PurpleAir sensors (PurpleAir Inc., Draper, UT, USA) deployed in triplicate in the Durango Complex in Phoenix, Arizona, USA during May 2019. The device or algorithm changes made by the manufacturer since this test may have changed the performance. Figure 3 shows the testing meta data dictionaries for organization information and testing location details. This information is used to populate testing reports.

Sensor data were read from three local csv files (assigned the unique serial identifiers PA01, PA03, and PA05) and we focused on the cf = atm estimates (PM_2.5_ and PM_10_) from channel A. The AirSensor.sensor_setup() method describes the contents of those files, indicates the SDFS parameters associated with each column, describes the timestamp format, specifies the time zone, and assigns each sensor a unique identifier derived from the serial number. A flowchart of this process is shown in Figure 2. Users who wish to analyze files from sensors of the same make, model, and firmware version for subsequent evaluations may use an existing setup.json file in the sensor_setup() module. Since the data formatting will be similar, this saves time by reducing the amount of manual entry required in order to instruct the setup routine on how to convert the sensor-specific data format into the SDFS data structure. For this report, we ran the reference_setup() module and the reference data was ingested locally. The Appendix A contains pages 1–4 of the report generated that evaluates the PM_2.5_ sensor performance.

On the first page of the report, the time-series (Figure 4) graph shows the PM_2.5_ pollutant concentrations reported by the reference FEM instrument and the three collocated sensors. The top graph shows the 1 h averages, and the bottom graph shows the 24 h averages. From these graphs, users can see that the sensors display the same trends as the reference instrument. Sensors usually under-report concentrations throughout the concentration range experienced in this test.

Scatterplots for individual sensors are included in the report and shown in Figure 5. These graphs indicate that the 1 h-averaged PM_2.5_ sensor concentrations have a moderately linear relationship with the reference FEM instrument (R^2^ = 0.64–0.66) and all sensors have similar error (RMSE = 4.42–4.94 µg/m^3^). Sensors generally under-report concentrations in this concentration range. The time-synced data points are colored based on the reference RH which is low in this geographical area (predominantly <50%). In Figure 5, a prominent two-prong relationship is observable, with the lower “arm” showing that the sensors frequently under-report the ambient concentration. This “arm” correlates with seasonal dust events. The dust events include both fine and coarse particles, which elevate the PM_2.5_ and PM_10_ concentrations. A further discussion of the sensor’s PM_10_ performance will be presented later in this section.

The first page of the report includes the performance metrics and target values/ranges graph shown in Figure 6. Potential users can quickly determine that these sensors meet the performance targets for the intercept (a portion of bias measurement), error (RMSE), and precision (CV and SD). These sensors nearly meet the target for the slope (another bias measurement) which may be addressed using a correction equation. Although the NRMSE error target is not met, this metric can be influenced by the concentration range experienced during the test so it can be deprioritized as the Sensor Targets Report suggests it is a more important metric as concentrations become higher (e.g., wildfire smoke conditions). The sensors do not meet the linearity (R^2^) target with 24 h-averaged data but are closer to the target at 1 h-averages (R^2^ = 0.64–0.66). Linearity improvements may be possible with more complex data processing algorithms to handle periods impacted by dust.

At the bottom of the first page are two pairs of graphs (Figure 7 and Figure 8) that investigate T and RH. Figure 7 summarizes the 1 h-averaged T and RH, as measured by the reference station’s meteorological instruments during the collocation period. Potential users should note that additional testing may be needed if these sensors will be used in colder climates (T < 24 °C or ~70 °F) or more humid climates (RH > 75%). This test cannot provide insight as to the performance or robustness of the sensors under those conditions. Figure 8 shows how the normalized (sensor/reference) PM_2.5_ concentrations change as a function of T and RH. In this case, there is little to no observable trend, suggesting RH may not largely impact the sensor performance.

Figure 9 shows similar 1 h-averaged scatterplots with the same sensor and reference data source but for PM_10_. The slope indicates that the sensors significantly under-report PM_10_ concentrations. With a weak linearity and high error, the sensors do not perform well in measuring PM_10_ and would not be a good choice for projects aimed at understanding dust exposure and sources.

The PA01 sensor plot from Figure 5 is shown again in Figure 10, but, this time, the dots are colored by the PM_2.5_/PM_10_ ratio as measured by a collocated reference instrument. This is an example of an exploratory plot that testers might find useful in further explaining some of the observations made during testing. Figure 10 was not created within *sensortoolkit* but is an example of how further analysis is made easier by starting with the reformatted data files and the calculated ratio created by the tool. The PM_2.5_/PM_10_ ratio can be used to better understand if there is variation in the particle size distribution. The “arm” that shows significant under-reporting has a lower PM_2.5_/PM_10_ ratio when compared to points that lie closer to the 1:1 line, indicating that higher concentrations of larger particles occur during this time period consistent with the observed dust events. As hypothesized previously, the sensor’s PM_2.5_ performance may be enhanced if shifts in the particle size distribution could be identified using only the sensor’s measurements.

## 4. Discussion

In the example presented, *sensortoolkit* was used to explore the “out-of-the-box” performance of the PurpleAir PA-II-SD sensor for the measurement of PM_2.5_ (Figure 4, Figure 5, Figure 6, Figure 7 and Figure 8 and Appendix A) and PM_10_ (Figure 9 and Figure 10 and Appendix A). This reflects the performance potential the user can expect of this sensor when purchased and deployed in an environment like the one in which it was tested (Phoenix, AZ, USA). The analysis concluded that the sensors were precise (SD = 0.2–0.3 μg/m^3^) and illustrated the trends in the PM_2.5_ concentration while under-reporting the ambient concentrations by almost half, indicating a need to develop and apply a correction to increase the comparability of this sensor’s data with FRM/FEM monitors. When compared to a collocated FEM, the comparative response was moderately linear (R^2^ = 0.64–0.66), and the agreement could be improved by isolating the lower arm from the scatterplot associated with seasonal dust effects. A review of the PM_10_ testing report further illustrates the sensor’s poor performance in the measuring of PM_10_, especially here in Phoenix where PM_10_ is dominated by PMc rather than PM_2.5_ (PM_2.5_/PM_10_ < 0.25), driven by dust. From these reports, prospective sensor users should learn that the PM_2.5_ data from this sensor will be biased low in the 0–40 μg/m^3^ range unless a correction is applied and that this sensor is not a robust choice for monitoring dust impacts or PM_10_, especially in locations where PMc may dominate. Because these reports were collected in Phoenix when RH < 50% and T > 15 °C, users should use caution when intending to collect data in environments which are more humid or colder, as these performance results may not hold. Indeed, other research confirms the humidity can impact the performance of this sensor [31,32,33,34,35].

At this time, it is common for data from PurpleAir sensors to be quality-assured to remove questionable data and to be bias-corrected [33,36] to make the data more comparable to FRM/FEM data. Because the *sensortoolkit* library was created solely to help testers create sensor evaluation reports, the library does not have the functionality to develop or apply sophisticated data correction models. However, users can apply corrections to the data before import or adapt the code by adding on additional modules. In fact, the EPA’s Sensor Targets Reports encourage testers to produce additional evaluation reports reflecting the recommended quality assurance and correction methodology for the sensor.

The example presented here is focused on PM (both the PM_2.5_ and PM_10_ reports for this sensor are provided in Appendix A). Although this is a PM example, the code library was designed to handle gases for which targets have been established including O_3_, NO_2_, CO, and SO_2_ [21,22,23,25]. The data ingestion module is sensor-agnostic and works with a wide variety of data formats. To illustrate the wide applicability of this tool, performance evaluation reports were produced using *sensortoolkit* for five ozone sensors and six PM_2.5_ sensors tested in one or more of seven locations within the U.S. Table 1 summarizes the testing locations, sensor types, and pollutants evaluated. The full reports are available in the Appendix A.

*Sensortoolkit* is currently limited in its application as a tool to generate a summary of the sensor performance evaluation results based on the EPA’s Sensor Targets Reports [21,22,23,25]. As such, it is limited in the pollutants and parameters it has been designed to handle (Appendix A), metrics calculated, and visualizations rendered. It was designed for sensor evaluators with Python experience. Some applications may require an expanded list of performance metrics or analyses, the ability to flag and/or remove specific data points, a more sophisticated statistical analysis of meteorological influences, and the ability to generate and apply a variety of data correction techniques. However, this Python code library is open-source, allowing future users to reuse and adapt the code to their particular uses by creating additional code modules. We encourage users to share their adapted code and use cases with others. The EPA does not currently anticipate making code updates or changes.

As mentioned in the introduction, other organizations have proposed solutions in response to the need for software tools that allow an analysis and visualization of the sensor data. Table 2 includes examples of tools which are publicly accessible (non-proprietary), open-source (code available), and modern (currently available), and have some functionality that enables the comparison of collocated devices (ideally, sensor data with a nearby reference monitor). *Sensortoolkit* is the only example written for Python users. Some tools can be used online, while others are meant to be downloaded and run from your computer. Although *sensortoolkit* does not have a graphical user interface, like some of the other tools listed, it does provide prompts that walk a user through a procedure to describe their data file. The code then reformats the data so that it can be quickly analyzed and visualized, unlike other tools which require the user to transform their own data into a standard format without assistance. *Sensortoolkit*’s robust data ingestion module works for a wide variety of data formats, whereas other tools are limited to a single or small subset of sensor types. Lastly, *sensortoolkit* is the only tool that will calculate all the metrics and create the complete set of visualizations requested in the EPA’s Sensor Targets Reports [21,22,23,25].

The ultimate goal in creating and offering the *sensortoolkit* Python code library is to make it easier for testers to process collocation data to ensure that sensor performance testing results are presented in a streamlined and interpretable manner with metrics that are defined and calculated consistently. Not only does this reduce the data-processing burden for testers, but the standard metrics and visualizations make it possible to compare performance across sensor types and evaluators. The U.S. EPA hopes that manufacturers will make it a goal to meet the performance targets and that these efforts will result in more robust and accurate sensors, providing benefits to sensor and sensor data users alike. As testers take up the recommendations made in the EPA’s Sensor Targets Reports, the U.S. EPA also hopes that it will become easier for potential users to find real-world evaluation results which provide the insights necessary to support gathering high-quality data for their intended application.

## 5. Conclusions

The U.S. EPA developed the *sensortoolkit* as a free, open-source tool to assist testers in generating a summary of results from sensor performance evaluations based on the Sensor Targets Reports. This article introduces the python code library and highlights the following:The user needs that guided the creation of this tool;A data ingestion methodology to handle variation in data format;Standardization in performance metric calculations;Data visualizations;The motivations for the reporting template design.

In this paper, we demonstrated the use of *sensortoolkit* to evaluate the out-of-the box performance of the PurpleAir PA-II-SD sensor in Phoenix, AZ. Although this is a brief glimpse at how *sensortoolkit* can streamline data processing and facilitate the presentation of the results, the U.S. EPA has used this tool to evaluate and describe the performance of at least six PM_2.5_ and five O_3_ sensors from different manufacturers, representing the variation in sensing elements, data processing, and performance in seven climatic regions of the U.S. These efforts illustrate the utility and versatility of this tool. As manufacturers take up the U.S. EPA’s recommendations, we hope to see more public reporting of test results. Of note is the Clarity Movement Co.’s early adoption, with several testing reports published on their website (https://www.clarity.io/collocation-results-library (accessed on 1 September 2025)).

We hope that consistency in reporting testing results will help standardize sensor performance evaluation approaches and encourage innovation and improvements in sensor technologies. Additionally, we hope that consumers can make more informed decisions when purchasing sensors by allowing them to compare the standard performance metrics and reports to find options that are the most appropriate for their application of interest, leading to more effective air quality monitoring.

## Figures and Tables

**Figure 1 sensors-25-05645-f001:**
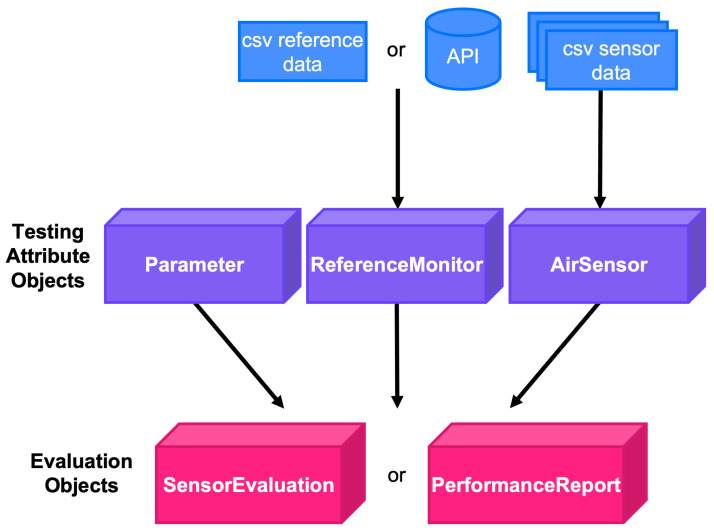
Schematic showing the *sensortoolkit* objects and general flow of information from data files to creation of the sensor performance report.

**Figure 2 sensors-25-05645-f002:**
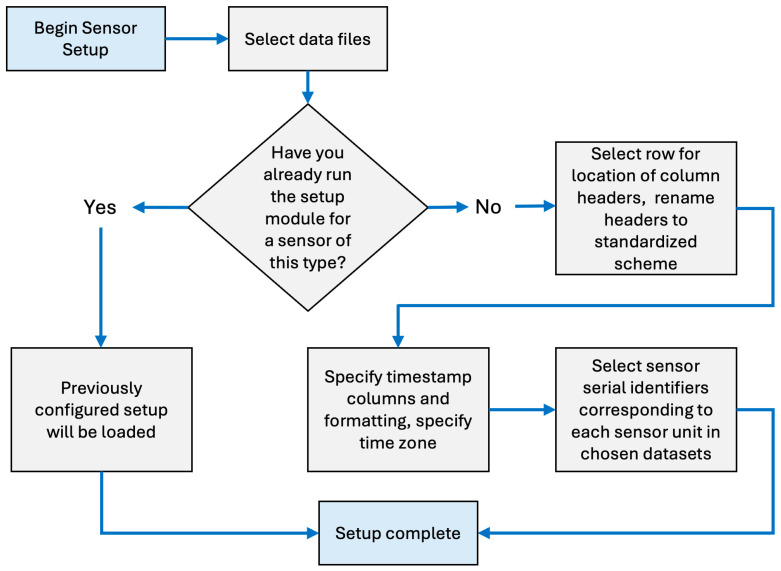
Flowchart of the choices and steps involved in the sensor_setup() module.

**Figure 3 sensors-25-05645-f003:**
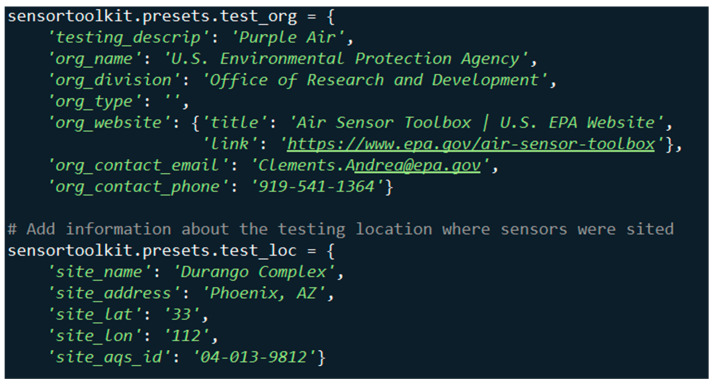
Meta data initially provided to *sensortoolkit* to be displayed on the report generated.

**Figure 4 sensors-25-05645-f004:**
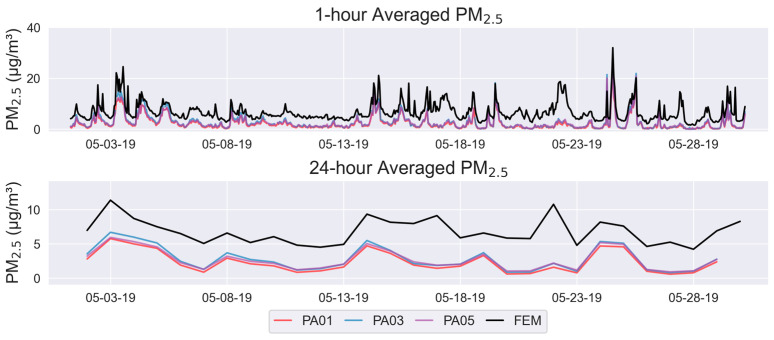
Time-series plots (1 h (**top**), and 2-h (**bottom**)) on Page 1 of the reporting template. Black lines represent the reference instrument, and colored lines represent the sensors.

**Figure 5 sensors-25-05645-f005:**
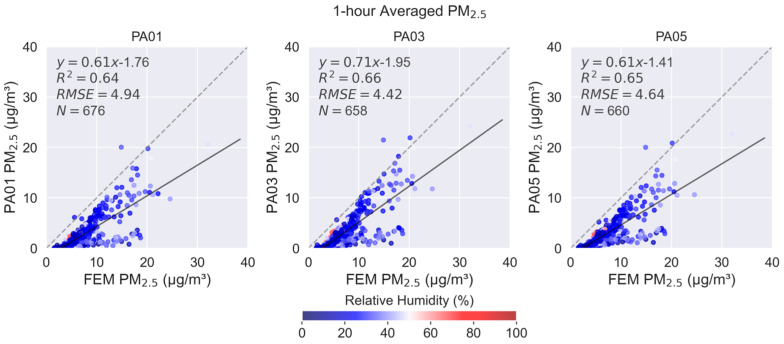
1 h-averaged PM_2.5_ concentration scatterplots for each of the three collocated sensors showing the linear regression (black line), 1:1 line (gray dotted line), and summary statistics (upper left of each graph). Dots are colored by relative humidity.

**Figure 6 sensors-25-05645-f006:**
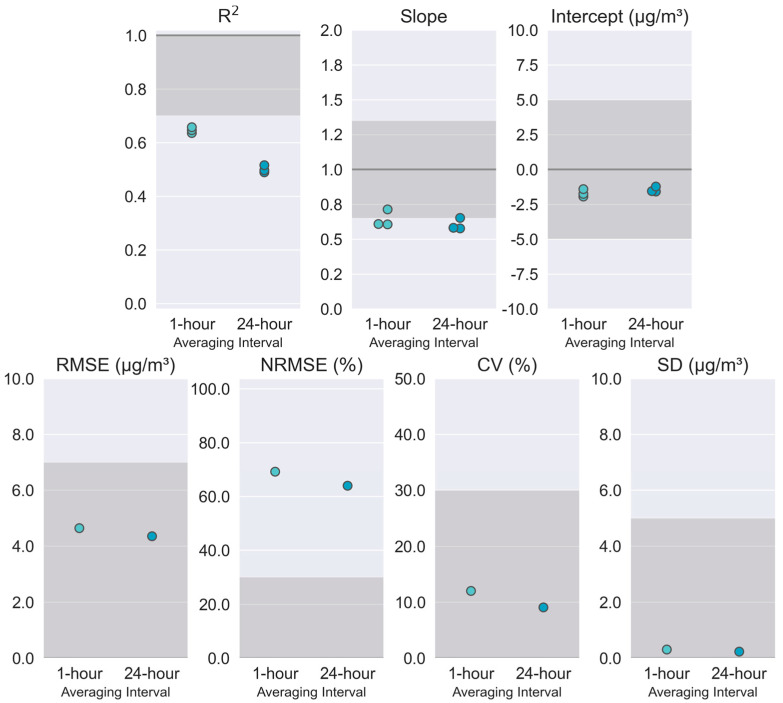
PM_2.5_ performance metrics (colored dots) and target value/range (darker grey shading) showing how the performance of these sensors compares to the targets at both 1 h and 24 h averages.

**Figure 7 sensors-25-05645-f007:**
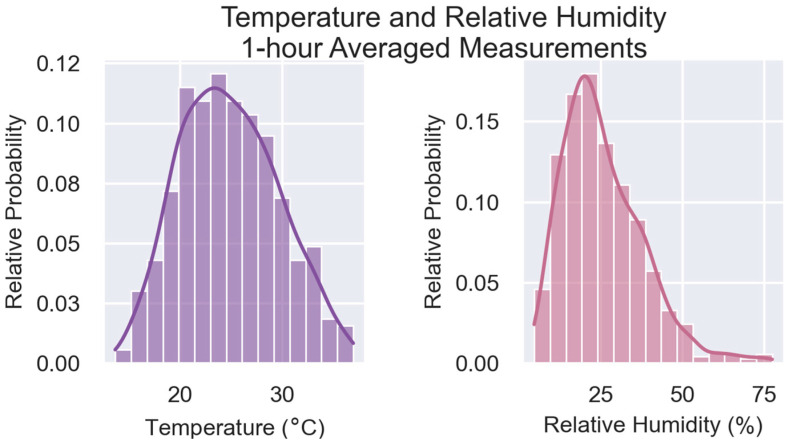
Relative probability of temperature (**left**) and relative humidity (**right**) recorded during the testing period are displayed as histograms. Measurements are grouped into 15 bins, and the frequency measurements within bin is normalized by the total number of measurements.

**Figure 8 sensors-25-05645-f008:**
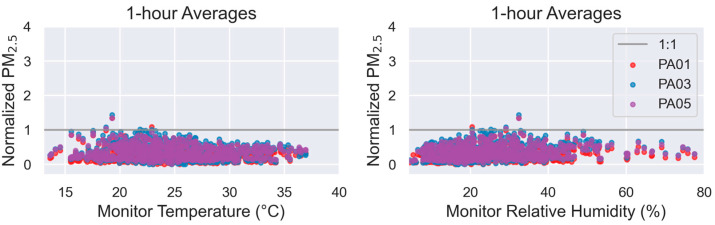
1 h sensor measurements divided by each hourly reference measurements (i.e., Normalized PM_2.5_) are shown for temperature (**left**) and relative humidity (**right**). Scatterplots for each sensor are displayed, as well as a gray 1:1 line that indicates ideal agreement between the sensor and reference measurements over the range of metrological conditions.

**Figure 9 sensors-25-05645-f009:**
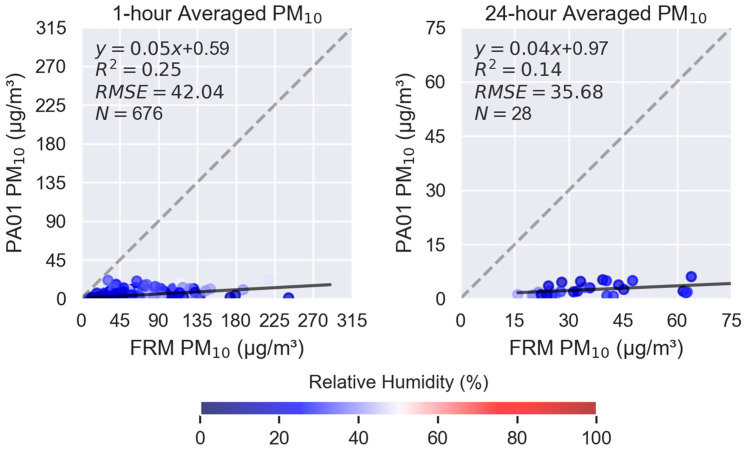
1 h-and 24-averaged PM_10_ concentration scatterplots for one of the collocated sensors showing the linear regression (black line), 1:1 line (gray dotted line), and summary statistics (upper left of each graph). Dots are colored by relative humidity.

**Figure 10 sensors-25-05645-f010:**
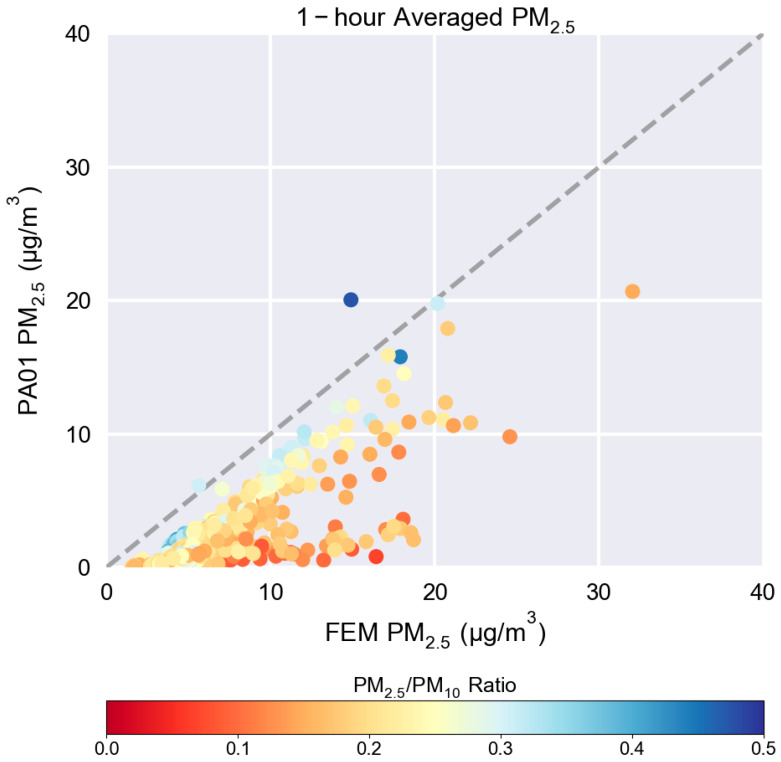
1 h-averaged PM_2.5_ concentration scatterplot for one of the collocated sensors showing the 1:1 line (gray dotted line) and plot markers colored by PM_2.5_/PM_10_ ratio.

**Table 1 sensors-25-05645-t001:** Example sensor performance reports included in the Appendix A.

Project	Location(s) Evaluated	Sensor(s) Evaluated	Pollutant(s) Evaluated
Phoenix	Phoenix, Arizona	PurpleAir PA-II-SD	PM_2.5_, PM_10_
Long-term performance project	Phoenix, Arizona Denver Colorado Wilmington, Delaware Decatur, Georgia Research Triangle Park, North Carolina Edmond, Oklahoma Milwaukee, Wisconsin	Aeroqual AQY	PM_2.5_, O_3_
		APT Maxima	PM_2.5_
		Clarity Node	PM_2.5_
		PurpleAir PA-II-SD	PM_2.5_
		SENSIT RAMP	PM_2.5_, O_3_
		QuantAQ AriSense	PM_2.5_
Research Triangle Park	Research Triangle Park, North Carolina	Apis APM01	O_3_
		Myriad Sensors PocketLab Air	O_3_
		Vaisala AQT420	O_3_

**Table 2 sensors-25-05645-t002:** The strengths and limitation of publicly available open-source data tools that may support the comparison of collocated sensor and reference monitor pairs.

Developer	Software Package of Library (Language)	Strengths	Limitations
Collaborative Development	aiRE (R/RShiny) [37]	▪ Enables data cleaning, analysis, visualization, and reporting	▪ Requires customization to meet the requirements of other countries ▪ Incomplete list of performance metrics and visualizations
Kings College London	OpenAir (R) [28]	▪ Robust data filtering and visualization options ▪ Robust documentation available to support newer R users	▪ User must format and import data as requested ▪ Lacks built-in data flagging and correction tools ▪ Requires R knowledge
South Coast Air Quality Management District	AirSensor (R) [26,27]	▪ Advanced sensor evaluation features (e.g., aging, drift, outliers)	▪ Limited to use with a short list of PM_2.5_ sensor types ▪ Incomplete list of performance metrics and visualizations
South Coast Air Quality Management District	Dataviewer (RShiny) [27]	▪ Web application ▪ Functionality of AirSensor	▪ Same as AirSensor
U.S. EPA	Air Sensor Data Unifier (RShiny) [38]	▪ Graphical user interface ▪ Data import and export into a variety of formats	▪ No visualizations of air quality or performance
U.S. EPA	Air Sensor Network Analysis Tool (RShiny) [39]	▪ Graphical user interface ▪ Data flagging and correction ▪ Plots and tables allowing users to understand sensor performance and local air quality ▪ Uses similar metrics to EPA’s performance targets	▪ User must format and import data as requested OR use the Air Sensor Data Unifier ▪ Does not prepare performance evaluation reports
U.S. EPA	RETIGO (browser-based) [40,41]	▪ Web-based graphical user interface ▪ Plots user data on a map and graphs allowing comparison with public data sources ▪ Calculates some similar metrics to EPA’s performance targets	▪ User must format data as requested ▪ Limits the number of files that can be uploaded ▪ Allows only simple linear correction applied to exported data
U.S. EPA	sensortoolkit (Python) [29]	▪ Data ingestion and formatting support for a wide variety of sensors ▪ Standardized metric calculation based on EPA’s performance targets ▪ Preparation of standard-format performance evaluation reports	▪ Lacks built-in data flagging and correction tools ▪ Limited to EPA’s criteria air pollutants ▪ Requires Python knowledge
U.S. EPA	SENTINEL (RShiny) [42]	▪ Graphical user interface ▪ Visualizations supporting sensors at the fenceline of industrial facilities or for emergency response ▪ Includes quality assurance and baseline correction	▪ Targets volatile organic compounds and methane sensors

## Data Availability

After publication, the U.S. EPA will make the data underlying this manuscript available at https://doi.org/10.23719/1531904. The data included in the example in the paper was collected as part of a larger research effort called Phoenix-as-a-Testbed for Air Quality sensors (P-TAQS). The larger dataset is available at https://doi.org/10.5281/zenodo.15174476 (accessed on 1 September 2025). The larger dataset for the long-term performance project is available at https://doi.org/10.5281/zenodo.14852865 (accessed on 1 September 2025).

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
