# Peer review of "Sensortoolkit—A Python Library for Standardizing the Ingestion, Analysis, and Reporting of Air Sensor Data for Performance Evaluation"

_sensors, 2025, doi:10.3390/s25185645_

Round 1

Reviewer 1 Report

Comments and Suggestions for Authors

Critical Analysis of the Article Structure

In the introduction, it is essential to incorporate a comparative study, presented in the form of a table or another appropriate structure. This study should discuss the strengths and areas for improvement of the proposed system compared to existing systems, such as the AirSensor R package [26], Mazama Science [27], and OpenAir [28].

The article lacks dedicated Conclusions and Future Work sections. It is recommended to:

Add a Conclusions section to summarise key findings, study limitations, and practical implications.

Include a Future Work section outlining:

  • Multi-site studies under varied climatic conditions to assess robustness.
  • Validation with gas sensors (O₃, NOâ‚‚, SOâ‚‚) in addition to particulate-matter sensors.
  • Comparative benchmarking across different sensor brands and models.
  • Development of region-specific correction algorithms.

Insufficiency in Tool Validation

The study presents a significant methodological limitation by restricting the validation of the sensortoolkit solely to a single case study using PurpleAir sensors. This approach substantially limits the demonstration of the tool’s applicability and robustness.

Recommendations:

  1. Sensor Diversification: Validate the toolkit with sensors from multiple manufacturers to demonstrate broad compatibility.
  2. Inter-Manufacturer Validation: Include comparative analyses across different sensing technologies to assess result consistency.
  3. Multi-Organizational Studies: Conduct collaborative validations with various institutions to verify reproducibility and operational reliability.

Points to improve in the article format:

  • Presentation of keywords: The keywords do not follow the journal's model and should be adapted to comply with the format required by the editorial guidelines.
  • In line 9, remove “mailto” of the mailto:kumar.menaka@epa.gov.
  • In line 42, coefficient of determination (R2) should be (R2).
  • "NRSME" appears in the text (line 450), which seems to be a typographical error of "NRMSE".
  • Citation format: Citations have inconsistent formatting as inappropriate spacing. Line 71
  • The chemical symbols NOâ‚‚ and SOâ‚‚ are not formatted correctly in the references section.
  • Abbreviations section: The abbreviations section is incomplete. For example, ISO, JSON, CSV, and CFR are missing.
  • Figure quality: In Figure 1, the size of text in the boxes should be increased to improve readability.
  • Problems with the website link for the AirNow-Tech website (lines 227-228).
  • In Figure 2's flowchart, the "Have you already run the setup module for sensor of this type?" block should be represented as a diamond (rhombus), not a rectangle.
  • Problems with DOI 23719/1531904 (line 539)
  • Regarding reference [3], the DOI is missing. “ Madhwal, S., et al. Evaluation of PM2.5 spatio-temporal variability and hotspot formation using low-cost sensors across urban-rural landscape in lucknow, India. Atmospheric Environment 2024, 319, p. 120302. DOI:”
  • "An inconsistency was observed in the spelling of the word 'flowchart/flow chart' throughout the document
  • Suggestions for Improving the Figures: It would be advisable to center Figure 7. Additionally, Figure 6 should be divided in half, creating two rows with graphs, to allow for better accuracy in reading the results and visualizing the axis labels.

Reviewer 2 Report

Comments and Suggestions for Authors

1.Although the article introduces the sensortoolkit library, the explanation of how the library handles data differences among various air sensor models is not detailed enough.

2.The article provides performance evaluation metrics (such as RMSE, R²) to measure sensor accuracy. However, it is recommended to conduct a more in-depth statistical analysis to explore the significance of performance differences among sensors, especially those related to meteorological factors. 

3.It is necessary to further discuss how to handle and adjust meteorological variables in different datasets. 

4.Providing more examples of actual deployment scenarios and discussing the challenges users may face would enhance the practicality of the toolkit. 

5.Although the article provides a detailed description of the toolkit's functions, it lacks a discussion of its limitations.

Round 2

Reviewer 2 Report

Comments and Suggestions for Authors

I have no problem for this version.